# Hemostatic Effects of Raloxifene in Ovariectomized Rats

**DOI:** 10.3390/life13071612

**Published:** 2023-07-23

**Authors:** Denys Alva-Chavarría, Maribel Soto-Núñez, Edgar Flores-Soto, Ruth Jaimez

**Affiliations:** Departamento de Farmacología, Facultad de Medicina, Universidad Nacional Autónoma de México, Mexico City 04510, Mexico; denysach@hotmail.com (D.A.-C.); farma_estrogenos@yahoo.com (M.S.-N.); edgarfloressoto@yahoo.com.mx (E.F.-S.)

**Keywords:** raloxifene, 17β-estradiol, hemostasis, coagulation, thrombosis

## Abstract

This study aimed to explore the effects of raloxifene (Rx) and estradiol (E_2_) on prothrombin time (PT), partial thromboplastin time (APTT), coagulation factors (VII, X, XI), and fibrinogen concentrations in rats. Female rats were ovariectomized 11 days prior to starting the treatment. Afterward, they received Rx or E_2_ (1, 10, 100, and 1000 µg/kg) or propylene glycol (0.3 mL; vehicle, V) subcutaneously for 3 consecutive days. Plasma was collected to measure the hemostatic parameters. Rx significantly increased PT (8%, at 1000 µg/kg; *p* < 0.05) and APTT at all doses evaluated (32, 70, 67, 30%; *p* < 0.05, respectively). Rx (1, 10, 100, and 1000 µg/kg) decreased the activity of factor VII by −20, −40, −37, and −17% (*p* < 0.05), respectively, and E_2_ increased it by 9, 34, 52, and 29%. Rx reduced factor X activity at 10 and 100 µg/kg doses (−30, and −30% *p* < 0.05), and E_2_ showed an increment of 24% with 1000 µg/kg dose only. Additionally, Rx (1, 10, 100 µg/kg) diminished FXI activity (−71, −62, −66; *p* < 0.05), E_2_ (1 and 10 µg/kg) in −60 and −38, respectively (*p* < 0.05), and Rx (1000 µg/kg) produced an increment of 29% (*p* < 0.05) in fibrinogen concentration, but not E_2_. Our findings suggest that raloxifene has a protective effect on hemostasis in rats.

## 1. Introduction

Raloxifene (Rx, Optruma^®^—Eli Lilly or Evista^®^—Daiichi Sankyo) currently has FDA approval to be used for the treatment and prophylaxis of postmenopausal osteoporosis [1]. Rx is a structural analog of tamoxifen and belongs to the second-generation selective benzothiophene group of selective estrogen receptor modulators (SERMs). Rx is a non-steroidal compound that acts as an ER ligand and has the unique characteristic of behaving as a selective agonist or antagonist depending on the function of certain tissues and the biological context in which they act. Like 17ꞵ-estradiol (E_2_), Rx is capable of crossing the cytoplasmic and nuclear membranes; once Rx is located in the nucleus, the benzothiophene ring can bind to the ER with an affinity equivalent to E_2_ [2,3]. The dual activity of the drug on the receptor is tissue-specific, either acting as an estrogenic agonist (bone, lipid metabolism) or as an antagonist (uterus and breast), attributing this quality to some of the beneficial effects related to these signaling pathways, reporting a low incidence and severity of adverse effects [2,3].

Rx is rapidly absorbed after oral administration and is extensively bound to plasma proteins, has a wide distribution, and is detected in most tissues minutes after ingestion, with the lowest concentration detected in the brain due to difficulty crossing the blood-brain barrier [4]. It undergoes first-pass effect and hepatic metabolism with the formation of glucuronide conjugates; finally, it is eliminated mainly in the feces [4]. Although there is a low incidence of reported side effects, some of the known adverse effects of Rx are antithrombotic and vascular actions, involving suppression of oxidative stress, vasorelaxation, and regulation of endothelial reaction to trauma, a predictive indicator for vascular disease [5,6,7]. Despite its beneficial effects on preventing bone loss and reducing fracture risk, Rx therapy is significantly associated with an increased risk (2- to -3-fold) of deep venous thrombosis and pulmonary embolism. This increment might be due to its estrogenic effects on the coagulation and fibrinolytic systems [8,9]. In one report, Rx therapy in postmenopausal women induced a procoagulant state, observing higher plasma levels of factor VIII, XI, and XII as well as diminished activated protein C (APC) sensitivity [10]. However, the mechanisms involved are not fully understood and further study is needed to explain Rx effects on the hemostasis and specific hemostatic parameters.

A useful model for the study of the effects of Raloxifene is through the use of ovariectomized (OVX) rats. The procedure of ovariectomizing a rat entails the surgical removal of the ovaries, thus removing the primary organ of estrogen biosynthesis, resulting in a reduction of the circulating levels of estradiol. Estrogens are involved in various physiological processes, and estradiol, as well as SERMs, can exert effects over distinct tissues such as bone, uterine, serum cholesterol, and adipose tissues as well as on the hematological system [5,11]; specifically, OVX has been shown to cause a prothrombotic state [5]. Therefore, many studies use this method to limit the effect of the internal production of estradiol when investigating the effects of these hormones on the coagulation system [11,12,13].

Estrogens are shown to be intricately involved in the regulation of the balance between the procoagulant and anticoagulant activity in the blood, and the use of hormone replacement therapy or treatment with SERMs has been shown to have diverse effects on this system [14]. It should be noted that the effects of raloxifene on the coagulation systems have mixed findings. In a randomized control trial of breast cancer prevention in postmenopausal women, treatment with Rx had a 3-fold increase in the risk for venous thrombosis [15]. Similarly, the current use of raloxifene in women with breast cancer was associated with a 7-fold increase in the risk for idiopathic venous thromboembolism in the absence of any other risk factors [16]. Moreover, in OVX pigs, Rx treatment increased the production of proaggregatory prostanoids [17].

On the other hand, Rx has also demonstrated beneficial homeostatic effects, preventing thrombotic events, as shown by Abu-Fanne et al. in OVX mice, where Rx reduced intravascular thrombosis and increased the expression of cyclooxygenase-2 and suppressing the adhesion of platelets [5]. Rx has also been suggested to reduce the risk of thrombosis through its direct effects on the vascular tissue. In aortic strips from OVX rats, Rx treatment has been shown to inhibit platelet aggregation, the direct effect on the vascular tissue increased the production of nitric oxide, preventing the formation of blood clots [13]. However, in another study, there was no change in NO production in OVX rats treated with Rx, but the levels of von Willebrand factor were lowered and plasminogen activator inhibitor 1 (PAI-1) was increased compared to the control group [12]. These discrepancies may be due to variations in study design, dosage regimens, species, and specific parameters assessed.

To explore that area, we evaluated the effects of Rx on female ovariectomized rats, and we compared them with the effects of 17ß-estradiol (E_2_), the classic agonist estrogenic compound.

## 2. Materials and Methods

### 2.1. Materials

Raloxifene (hydrochloride [6-hydroxy-2-(4-hydroxyphenyl)-1-benzothiophen-3-yl]-[4-(2-piperidin-1-ylethoxy) phenyl] methanone), 17ß-estradiol (E_2_) (1,3,5(10)-estratrien-3,17ß-diol), tert-amyl alcohol, 2,2,2-tribromoethanol, and propylene glycol were purchased from Sigma-Aldrich (St. Louis, MO, USA). Thromboplastin C plus, Actin FS, coagulation factors deficient plasmas (FVII, FX, FXI), bovine thrombin, standard human plasma, veronal, and imidazole buffer solution were purchased from Dade^®^ Behring^®^ (Marburg, Germany). Vacutainer blood collection tubes with buffer sodium citrate and Safety-Lok^TM^ collection sets were purchased from Becton, Dickinson, and Co. (Franklin Lakes, NJ, USA).

### 2.2. Animals

The experiments were carried out under the approval of the ethics and research committee of the Faculty of Medicine of the National Autonomous University of Mexico and the standards of the official Mexican norm (NOM-062-ZOO-1999). Adult Wistar rats (Rattus Norvegicus) weighing 200 to 225 g from the Faculty of Medicine’s vivarium were used. The animals were kept under controlled light and dark conditions in cycles of 12 h, a humidity of 40%, and a temperature of 22 °C to 24 °C, and were kept in polypropylene boxes (4 rats per box). They received purified water and lab diet 5001. All testing was made in rigorous accordance with the international regulations of the National Institutes of Health Guide for the Care and Use of Laboratory Animals (NIH publication No. 80–23 revised in 1996).

### 2.3. Ovariectomy

The rats were ovariectomized (surgical removal of the ovaries), implying that they are in a state of estrogen deficiency. Ovariectomy models have significantly contributed to our understanding of the role of raloxifene and estrogen in the context of hemostasis and thrombosis [12]. Rats were ovariectomized under anesthesia with 0.2 g/kg tribromoethanol/tert-amyl alcohol, through intraperitoneal administration. After ovarian tubal ligation, the ovaries were removed from the uterine horns by a dorsal section, the uteri were tied with chromic gut surgical suture 4–0 and the dorsal incisions were closed. The animals were placed in clean cages in a warm environment for recovery.

### 2.4. Experimental Design

A total of 135 animals were used for the experiment, after 11 days of recovery, ovariectomized rats were distributed into nine groups according to their body weight. Due to the enclosure capacities, the experiments could only be carried out on nine groups at a time; four rats were assigned to each group. The first group was the control group which received the vehicle propylene glycol (0.3 mL, *n* = 15). The second to fifth group received Rx (total *n* = 60; 1, 10, 100, or 10,000 µg/kg). The dose of raloxifene used in a clinical setting for the treatment of postmenopausal women is 60 mg daily, which corresponds to a dosage of 1 mg/kg. For this reason, we based the study on this concentration and a range of lower concentrations from 1, 10, 100, and 1000 µg/kg in rats in an acute 3-day administration to detect if the treatment with raloxifene produced acute changes in the hemostatic markers [18,19,20,21]. The third group to the sixth to ninth group received E_2_ (total *n* = 60; 1, 10, 100, or 1000 µg/kg). All doses were administered subcutaneously for 3 consecutive days and the experiment was performed in triplicate.

### 2.5. Blood and Plasma Collection

In research studies involving ovariectomized rats, blood and plasma collection are common procedures performed to analyze hematological parameters [12]. A day after the last injection, the rats in all groups fasted overnight with ad libitum access to water, then they were anesthetized and euthanized with Avertin (ip, 250 mg/kg, tert-amyl alcohol/2,2,2-tribromoethanol) through intraperitoneal administration. Blood samples were collected from the iliac artery using a Vacutainer^®^ system into citrated (0.105 M) tubes (Dickinson and Company, Franklin Lakes, NJ, USA) as an anticoagulant. Each sample was centrifuged at 20–22 °C for 10 min at 800× *g* (3000 rpm) and the plasma was separated. All samples were stored at −20 °C until assays. It’s important to follow appropriate animal handling and ethical guidelines while performing blood and plasma collection in ovariectomized rats. This includes using aseptic techniques, ensuring proper anesthesia, and minimizing stress or discomfort to the animals.

### 2.6. Determination of Prothrombin Time (PT) and Activated Partial Thromboplastin Time (APTT)

Prothrombin time (PT) and activated partial thromboplastin time (APTT) are two common laboratory tests used to evaluate the clotting ability of the blood and assess the functionality of the coagulation system. These tests are standard tools useful for the diagnosis and monitoring of various bleeding and clotting disorders. PT and APTT were determined in a semiautomatic coagulometer, according to Quick [22], and Proctor and Rapaport methods [23], respectively. To activate PT, we added thromboplastin C plus. Samples (50 µL) were incubated for 60 s at 37 °C, then thromboplastin C plus (37 °C) was added. For APTT determinations, 50 µL of the sample was added and mixed with actin (50 µL) and incubated for 120 s at 37 °C. The samples were activated by adding CaCl_2_ (50 µL).

The standardization of coagulation factors VII, X, and XI was conducted in accordance with García-Manzano [24]. Briefly, samples (50 µL) were incubated (37 °C), with the plasma free of coagulation factors (50 µL) (VII, X, or XI). Samples were incubated at 37 °C for 60 s, and thromboplastin (50 µL) was added to activate clotting in factors VII and X. For factor XI determination, samples were mixed with actin (50 µL) and incubated at 37 °C for 120 s and clotting was activated with the addition of CaCl_2_ (50 µL). It’s worth noting that the PT and APTT tests are often performed together, along with other coagulation tests, to provide a comprehensive evaluation of the clotting system.

### 2.7. Standard Curve of Fibrinogen

The determination of fibrinogen levels in a blood sample can be performed using the Clauss method [25]. This method is widely used and established for fibrinogen determination. It involves the measurement of the clotting time of fibrinogen in the presence of excess thrombin. The principle is based on the conversion of fibrinogen to fibrin by thrombin, resulting in the formation of a clot. The clotting time is inversely proportional to the fibrinogen concentration in the sample. In the Clauss method, a blood sample is collected in a citrate anticoagulant tube to prevent clotting. The plasma is then separated by centrifugation. To initiate clot formation, a standardized thrombin reagent is added to the plasma. The clotting time is measured using a coagulation analyzer or a manual method, and it is compared to a calibration curve or standard to determine the fibrinogen concentration. Plasma concentrations (mg/dl) were prepared with a veronal buffer as follows: 500 (4.4 s), 375 (5.9 s), 250 (8.4 s), 187 (9.1 s), and 125 (23.7 s). Each sample was incubated for 120 s at 37 °C and activated with diluted thrombin 1:2 (100 IU) [25].

### 2.8. Effects of Raloxifene and Estradiol on Blood Coagulation Factors VII, X, and XI

To establish if both raloxifene and estradiol could have an effect on the blood coagulation factors VII, X, and XI in ovariectomized rats, you would typically follow a research study design specific to this animal model. The ovariectomized rats were randomly divided into different treatment groups. This included a control group (placebo or untreated), a raloxifene-treated group, and an estradiol-treated group. After the respective treatments with Rx and E2, the blood samples were obtained and centrifuged to obtain the plasma. Each sample was diluted with imidazole buffer. The dilution with imidazole buffer for factors VII and X (extrinsic and common pathway) was 1:20, whereas samples for factor XI determinations (intrinsic pathway) were diluted with imidazole buffer at 1:5.

To determine FVII and X activities, the samples (50 µL) were incubated at 37 °C with plasma, free of FVII or FX, for 60 s. To activate each factor, thromboplastin was added (50 µL). For the determination of FXI activity, we used samples treated with the compounds. Each sample (50 µL) was mixed with factor XI deficient plasma (50 µL) and actin (50 µL) and incubated for 120 s. The reaction was activated with the addition of prewarmed CaCl_2_ (100 µL). All data obtained were extrapolated in the standard curve of each factor. Vehicle data were considered 100%. The effects on coagulation factors should be considered in the context of overall hemostasis and thrombotic risk.

### 2.9. Determination of the Effects of Raloxifene and Estradiol on Fibrinogen Concentrations

The effects of raloxifene and estradiol on fibrinogen concentrations in ovariectomized rats can be investigated through experimental studies. The ovariectomized rats were randomly divided into different treatment groups. This included a control group (placebo or untreated), a raloxifene-treated group, and an estradiol-treated group. Fibrinogen was quantified with samples diluted 1:10 through the Clauss method [25]. Samples were incubated for 120 s at 37 °C and activated with 100 µL of thrombin 1:2 (100 IU/mL). The data obtained in seconds were extrapolated to obtain the concentration.

### 2.10. Statistical Methods

The differences in the response between the treated and control groups were evaluated through parametric tests: a one-way analysis of variance (ANOVA) followed by Dunnett’s test ANOVA. Statistical significance was set at *p* values lower than 0.05 (*p* < 0.05). All data were presented as mean or percentage ± standard error of the mean (SEM). The vehicle was considered 100%. In all experiments, the value for *n* “15” per concentration corresponds to a different animal, with a final *n* = 60 per group, and all experiments were conducted in triplicate and analyzed with SigmaPlot 10 (Systat Software, San Jose, CA, USA). 

## 3. Results

### 3.1. Raloxifene and Estradiol Effects on PT and APTT Screening Tests

We determined the effect of raloxifene and estradiol on the PT (prothrombin time) and APTT (activated partial thromboplastin time) screening tests, which are commonly used to evaluate the coagulation status of an individual. To evaluate differences between PT and APTT due to Rx and E_2_, we measured these parameters in blood serum. Figure 1A shows how the administration of Rx (1000 µg/kg) significantly increased PT (8%; *p* < 0.05), while the administration of E_2_ had no significant effect. Figure 1B shows that Rx (1, 10, 100, and 1000 µg/kg) produced a significant increment (*p* < 0.05) in APTT (32, 70, 67, and 30%, respectively), whereas E_2_ did not show any effect on this parameter. Rx showed an important increase in APTT, which is a marker of the intrinsic pathway of coagulation. It’s important to note that the effects of raloxifene and estradiol on PT and APTT tests can be influenced by many factors, including the dosage and duration of treatment.

### 3.2. Raloxifene and Estradiol Effects on Extrinsic Factor VII and Common Pathway Factor X

Both raloxifene and estradiol can have effects on extrinsic factor VII and common pathway factor X, which are important components of the coagulation cascade. Figure 2A shows that Rx administration (1, 10, 100, and 1000 µg/kg) significantly decreased the activity of factor VII by −20, −40, −37, and −17% (*p* < 0.05), respectively; whereas the administration of E_2_ produced a significant increment of 9, 34, 52, and 29% with all doses evaluated. Figure 2B shows that administration of 10 and 100 µg/kg doses of Rx (*p* < 0.05) produced a significant decrease of the activity of factor X by −30%, while a 1000 µg/kg dose of E_2_ showed an increment of 24%. Rx decreased the activity of both factors, as opposed to the effect with E_2_; however, their effects may differ due to their distinct mechanisms of action and interactions with estrogen receptors.

### 3.3. Raloxifene and Estradiol Effects on Intrinsic Factor XI and Fibrinogen Concentration

Studies investigating the effects of raloxifene and estradiol on intrinsic factor XI and fibrinogen in rats that have undergone ovariectomy (OVX) are limited. Our results, shown in Figure 3A, show Rx and E_2_ effects on coagulation factor XI. The administration of Rx (1, 10, 100, and 1000 µg/kg) produced a diminution in factor XI activity (−71, −62, −66%; *p* < 0.05), whereas E_2_ administration (1 and 10 µg/Kg) diminished factor XI activity in −60 and −38%, respectively (*p* < 0.05). In this case, both compounds diminished intrinsic factor XI. Figure 3B shows the effect of Rx and E_2_ on fibrinogen concentrations. The administration of Rx (1000 µg/kg) produced a significant increment of 29% (*p* < 0.05) in fibrinogen concentration. The administration of E_2_ produced a modest yet not significant decrease in fibrinogen concentration.

### 3.4. This Diagram Illustrates Our Current Understanding of the Effect of Raloxifene and Estrogen on Hemostasis and Thrombosis

The squeme represents the main effects of Raloxifene and estradiol on extrinsec, intrinsic and common pathways of coagulation. The objective of Figure 4 is to concentrate the information from all the figures.

## 4. Discussion

This study reveals that Rx, a specific type of estrogen receptor modulator, affects three different hemostasis pathways in rats: the extrinsic, intrinsic, and common pathways, as observed by the significant increment in both the prothrombin time and activated partial thromboplastin time. This effect of raloxifene was caused by regulating the activity of coagulation factors VII and XI while increasing fibrinogen concentration. Raloxifene modifies the coagulation parameters and decreases the risk of thrombosis in ovariectomized rats thereby showcasing its potential as a therapeutic option for conditions such as osteoporosis, especially in patients with a risk of venous thrombosis.

When it comes to the extrinsic pathway, the highest dosage of Rx led to an increase in prothrombin time and a notable decrease in factor VII activity. It could especially benefit patients at risk of venous thromboembolism as an alternative to estradiol and merits further study. On the other hand, E_2_ resulted in an increase in factor VII activity. Even though the understanding of the effects of estrogen on the hemostasis and thrombosis systems is still incomplete, it has been shown to affect multiple variables that influence the balance between the procoagulant and anticoagulant properties. The effects of estrogen should especially be taken into consideration in patients that have other risk factors for thrombosis (e.g., age, smoking, obesity, comorbidities, and genetic predisposition), especially when considering hormonal replacement therapy, and perhaps could benefit from a SERM alternative [26].

According to Roqué et al., raloxifene (Rx) had varying effects on hemostasis and thrombotic risk in postmenopausal women with ischemic heart disease. The study found that mid-term treatment with Rx did not have a significant effect on endothelial function in this group of women [27].

Interestingly, the effects of Rx seem to be associated with a chronological influence. Postmenopausal women treated daily with 60 mg of Rx in the morning for 12 months had higher plasma concentrations of plasminogen-activated inhibitor (PAI)-1 in comparison with the group receiving the evening treatment. No additional changes were observed in other coagulation factors. It should be noted that elevated PAI-1 levels are associated with a risk of venous thromboembolism and should be considered when prescribing Rx [28]. In another 12-week study on hormone therapy with either Rx or tibolone in postmenopausal women, the group with a history of venous thrombosis showed an increase in sex hormone binding globulin (SHBG), which in turn was associated with a change in APC resistance, proposing that SHBG can be used as a biomarker to identify an increased risk in venous thrombosis in hormone therapy [29].

Regarding hemostasis, the study showed that Rx affected the extrinsic, intrinsic, and common pathways in rats. In particular, the highest dose of Rx produced an increment in prothrombin time and a significant inhibitory effect in factor VII activity, which is associated with increased thrombotic risk. However, the same study reported that Rx treatment did not have any effect on fibrinolytic activity, which is involved in the breakdown of blood clots and can help prevent thrombotic events. It is important to note that our findings with Raloxifene had similar results. In our study with rats, we also observed that Raloxifene had varying effects on hemostasis and thrombotic risk. Specifically, Raloxifene had an increase in prothrombin time (at the highest dose of 1000 µg/kg) and significantly inhibits the activity of factor VII (highest effect at 10 and 100 µg/kg), suggesting an increase in thrombotic risk. However, we did not find any significant effect of Raloxifene on fibrinolytic activity, which is involved in the breakdown of blood clots and can help prevent thrombotic events [27].

It has been found that there are distinct differences in the effects of raloxifene and estrogen on coagulation factors, indicating that plasma levels of coagulation factors are regulated by independent mechanisms. One significant difference between raloxifene and estrogen is that only raloxifene decreased the levels of coagulation factors VII, X, and XI. In the intrinsic pathway, raloxifene led to a notable increase in activated partial thromboplastin time. Both raloxifene and estrogen reduced the activity of factor XI, but only at low doses. In terms of the common pathway, raloxifene decreased the activity of factor X, whereas estrogen showed the opposite effect. These findings suggest that raloxifene and estrogen have varying effects on different coagulation factors and pathways, highlighting the importance of careful consideration when prescribing these medications for patients with a history of thrombotic events or other relevant medical conditions [30].

Only the highest dose of Rx produced a significant increase in fibrinogen concentrations. Fibrinogen is an independent risk factor of initial cardiovascular events related to venous and arterial thrombosis [31,32]. It is important to note that the exact hemostatic effects of raloxifene may vary depending on the dosage, duration of treatment, and individual characteristics of the experimental models. Interestingly, estrogens, as do sex hormones in general, have a biphasic behavior dependent on the concentration used [33,34,35].

In 2001, Cushman reported that Rx appears to have more positive effects than hormone replacement therapy in lowering fibrinogen [36]. Therefore, when Rx is prescribed to postmenopausal women with osteoporosis, it is fundamental to take into consideration the dosing times in order for plasma levels not to increase to a concentration where the cardioprotective effect is lost [37]. Our findings support past reports in mice that stated that 4 months of treatment with Rx attenuates intravascular thrombosis. They showed that the antithrombotic effect was associated with an improved expression of cyclooxygenase-2 (COX2) and inhibition of platelet surface adhesion [5].

In contrast to our findings, Cosman’s research demonstrated that estrogen, tamoxifen, and raloxifene have pro-coagulation effects and impair anticoagulation in postmenopausal women, differing subtly from the effects of estrogen. Overall, our findings suggest that raloxifene can positively impact coagulation markers related to cardiovascular risk, and we observed a protective effect of raloxifene in the acute treatment of hemostasis in rats.

## 5. Conclusions

This study provides evidence that raloxifene has hemostatic effects on ovariectomized rats by influencing key factors in both the extrinsic and intrinsic pathways of the coagulation system. However, further studies are necessary to elucidate the underlying mechanisms of Rx’s hemostatic effects and to evaluate its safety profile in human subjects.

## Figures and Tables

**Figure 1 life-13-01612-f001:**
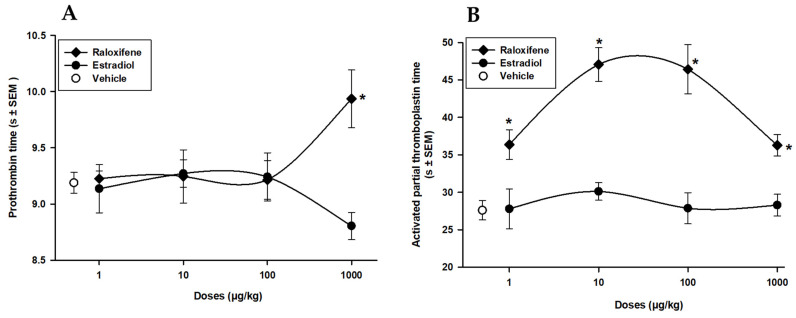
Raloxifene (⧫) and estradiol (•) on prothrombin (**A**) and thromboplastin (**B**) time. Each point represents the mean ± SEM of 15 determinations. * *p* < 0.05 vs. vehicle (o) Dunnett’s test.

**Figure 2 life-13-01612-f002:**
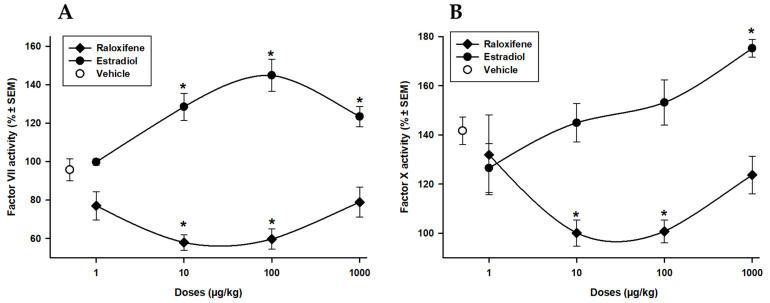
Raloxifene (⧫) and estradiol (•) on the activity of coagulation factors VII (**A**) and X (**B**). Each point represents the percentage ± SEM of 15 determinations. * *p* < 0.05 vs. vehicle (o) Dunnett´s test.

**Figure 3 life-13-01612-f003:**
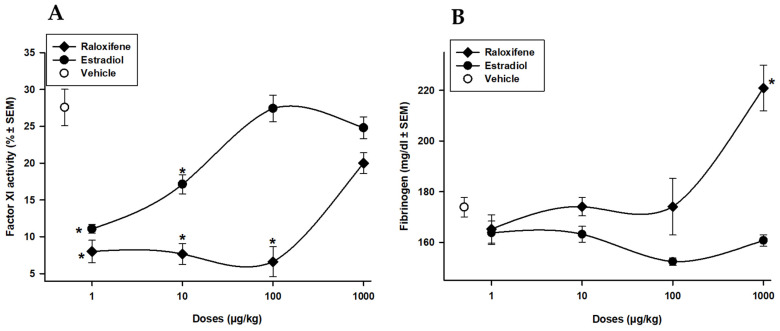
Raloxifene (⧫) and estradiol (•) on the activity of coagulation factor XI (**A**) and fibrinogen concentrations (**B**). Each point represents the percentage ± SEM of 15 determinations. * *p* < 0.05 vs. vehicle (o) Dunnett´s test.

**Figure 4 life-13-01612-f004:**
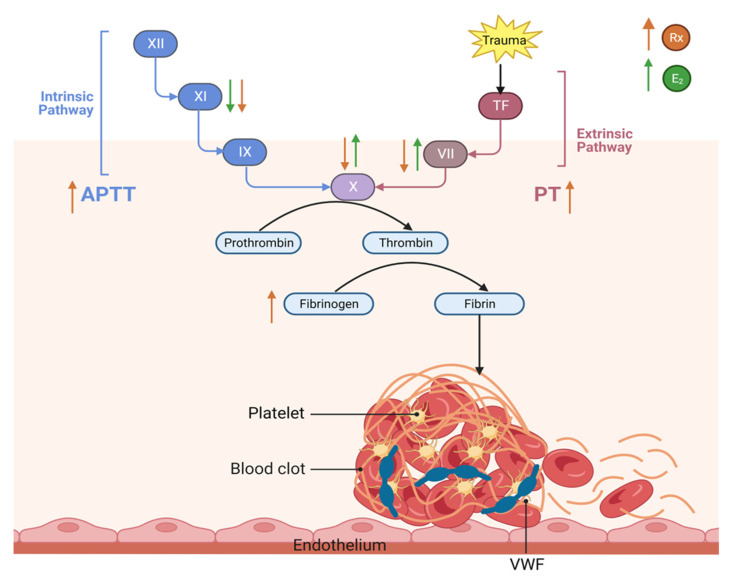
Raloxifene and estradiol have contrasting effects on hemostasis and thrombosis due to their interactions with estrogen receptors and their impact on various components of the coagulation system. The orange arrows represent the effect of raloxifene while the green arrows represent the effect of estradiol. Raloxifene showed an increased-on PT, APTT, and fibrinogen and produced a diminution in factor VII, X, and XI activity. E2 induced an increment of factor VII and X activity and lowered factor XI, whereas it did not show any effect on PT and APTT.

## Data Availability

The data presented in this study are available in this article.

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
