# Peer review of "Hemostatic Effects of Raloxifene in Ovariectomized Rats"

_life, 2023, doi:10.3390/life13071612_

Round 1

Reviewer 1 Report

The manuscript titled: Hemostatic effects of raloxifene in ovariectomized rats, sheds light on topic of effects of raloxifene (Rx) and estradiol (E2) on hemostatic parameters of experimental rats which represents a crucial in animal biology and hematology. Indeed, the idea of the manuscript is novel and it is well written using the standard format of life journal. However, the number of rats in this study is very low which limits the findings of this investigation reduces the quality of this current manuscript.

The package of data presented is also very low to get significant findings out of this investigation.

Therefore, I recommend the major revision of the current manuscript titled: Hemostatic effects of raloxifene in ovariectomized rats, sheds light on topic of effects of raloxifene (Rx) and estradiol (E2) on hemostatic parameters of experimental rats.  

Author Response

Reviewer 1

The manuscript titled: Hemostatic effects of raloxifene in ovariectomized rats, sheds light on topic of effects of raloxifene (Rx) and estradiol (E2) on hemostatic parameters of experimental rats which represents a crucial in animal biology and hematology. Indeed, the idea of the manuscript is novel and it is well written using the standard format of life journal. However, the number of rats in this study is very low which limits the findings of this investigation reduces the quality of this current manuscript.

The package of data presented is also very low to get significant findings out of this investigation.

A: Thank you for your comment, we believe there was a confusion in the way we explained our methodology, for each concentration there was an n=15 for each group by triplicate; therefore, for each group i.e., raloxifene for the 4 concentrations used there would be an n=60 by triplicate, and this is the case for all of the experiments. We have added the following text in the manuscript to clarify:

L:207-208  “per concentration” and “with a final n=60 per group”

Therefore, I recommend the major revision of the current manuscript titled: Hemostatic effects of raloxifene in ovariectomized rats, sheds light on topic of effects of raloxifene (Rx) and estradiol (E2) on hemostatic parameters of experimental rats. 

In this study, authors evaluated the hemostatic effects of Rx (with different doses) on female ovariectomized rats (compared them with the effects of 17ß-estradiol, E2). In addition to the available results,

A: Thank you for the kind suggestion, we have searched for the mentioned reference, however, the only thing we could find was a preprint of the present manuscript, please confirm if this is the suggestion reference. https://www.preprints.org/manuscript/202306.2215/v1.

Evaluation Report

Manuscript Title:

Title: Hemostatic effects of raloxifene in ovariectomized rats

Journal

Life

Manuscript ID

life-2470867

Section

Comments and recommendation

General comment

The manuscript titled: Hemostatic effects of raloxifene in ovariectomized rats, sheds light on topic of effects of raloxifene (Rx) and estradiol (E2) on hemostatic parameters of experimental rats which represents a crucial in animal biology and hematology. Indeed, the idea of the manuscript is novel and it is well written using the standard format of life journal. However, the number of rats in this study is very low which limits the findings of this investigation reduces the quality of this current manuscript.

The package of data presented is also very low to get significant findings out of this investigation.

Abstract

1-The abstract is well described and contained all required information of the study with clear objectives.

Introduction

1-The introduction section is well written and described the well the state of the art of this work problem. In addition, the authors have used up to date research articles relevant to the study topic.

Materials and methods

The design of this research work fits well with the aim and scope of this type of investigations. The methodologies used in are in accordance with type of work. In addition, the authors have used suitable statistical analysis of manuscript data.

However, the number of rats in this study is very low which limits the findings of this investigation reduces the quality of this current manuscript.

Results

1-The results of this manuscript are described the findings of the manuscript. However, the package of data presented is also very low to get significant findings out of this investigation.

Discussion

The results are well discussed in the discussion section using up to date literature which closely related to the study topic. However, the discussion inclusion of important data related to the histological findings and integration of ultrasound and histological data.

Conclusions

The conclusions of this manuscript described well the manuscript findings.

Bibliography/References

The list of references is well formatted according to journal instructions. Most of references listed in the manuscript are within 2002-2013 and closely relative to the research focus of study.

Recommendation and final comment

I recommend the major revision of the current manuscript titled: Hemostatic effects of raloxifene in ovariectomized rats, sheds light on topic of effects of raloxifene (Rx) and estradiol (E2) on hemostatic parameters of experimental rats.  The evaluation is based on that overall value of data presented and novelty of the idea in addition to the experimental design and quality of data presented in this manuscript writing. In addition, it is on the scope of this outstanding journal. However, the number of rats in this study is very low which limits the findings of this investigation reduces the quality of this current manuscript.The package of data presented is also very low to get significant findings out of this investigation.

We thank the reviewer’s comments, they have contributed greatly to our manuscript.

Reviewer 2 Report

In this study, authors evaluated the hemostatic effects of Rx (with different doses) on female ovariectomized rats (compared them with the effects of 17ß-estradiol, E2). In addition to the available results,

The authors need to add in the Introduction or Methods why they chose such a range (1,10,100 and 1000) of treatment doses. Meanwhile, the authors should analyze in the Discussion section whether the changes in the relevant indicators are intrinsically linked between the Rx and E2 treatment groups and why; furthermore, the magnitude of the change in dose 10, 100 seems to be more significant in the changes in some indicators (factor XI and factor VII activity) than in dose 1000, and why this is happened if there is no absolute dose-dependent linear relationship in the changes in these indicators along with doses, these are missing in the manuscript and need to be analyzed in the Discussion section.

Minors:

L11/L14: ‘1 to 1000 μg/Kg’, the specific experimental dose needs to be shown here, rather than using a concentration range.

L17: ‘1-100’, it means from 1 to 100, but only 1 and 100 dose were detected in this study, so ‘1, 100’ would be more accurate.

L50: APC, The full name is needed when the abbreviation is first shown.

L63: ‘a the’, please check the grammar.

L105: ‘lab diet 5001.’ What’s the mean 5001 here, please check and revise it.

L201: ‘as percentage ± standard error of the mean (SEM)’, In the context, there was presented as mean ± SEM (L228, L243). Please check and correct it.

L227,242, 263: ‘Raloxifene an estradiol’, ‘an’ or ‘and’, please check and revise them.

L315-317, ‘Specifically, the highest dose of Raloxifene resulted in an increase in prothrombin time and a significant inhibitory effect in factor VII activity’. Actually, for the inhibitory effect of Raloxifene dose in factor VII activity, the dose 100 and 10 has the highest effects (Fig2A).

Figures:

Figures 2: Panel B, Check the panel letters in the Fig2 and Fig3.  

L263, ‘Figure 3. Raloxifene an estradiol on activity of coagulation factor VII (A)’, In y axis of panel Fig3A, it was factor XI. Please check it.

NA

Author Response

Reviewer 2

The authors need to add in the Introduction or Methods why they chose such a range (1,10,100 and 1000) of treatment doses. Meanwhile, the authors should analyze in the Discussion section whether the changes in the relevant indicators are intrinsically linked between the Rx and E2 treatment groups and why; furthermore, the magnitude of the change in dose 10, 100 seems to be more significant in the changes in some indicators (factor XI and factor VII activity) than in dose 1000, and why this is happened if there is no absolute dose-dependent linear relationship in the changes in these indicators along with doses, these are missing in the manuscript and need to be analyzed in the Discussion section.

The authors need to add in the Introduction or Methods why they chose such a range (1,10,100 and 1000) of treatment doses

A: Thank you for your comment, the dose of raloxifene used in a clinical setting for the treatment of postmenopausal women is of 60 mg daily, which corresponds to a dosage of 1 mg/kg, for this reason we based the study in this concentration and a range of lower concentration from 1, 10, 100 and 1000 µg/kg in rats in an acute 3 day administration to detect if the treatment with raloxifene produced acute changes in the hemostatic markers. Additionally, in breast cancer prevention the therapeutic dosage can be up to 150 mg/kg. We have consulted the DrugBank, international guidelines as well as other animal studies with a similar group of study for our dosage range:

L:122 The dose of raloxifene used in a clinical setting for the treatment of postmenopausal women is of 60 mg daily, which corresponds to a dosage of 1 mg/kg, for this reason we based the study in this concentration and a range of lower concentration from 1, 10, 100 and 1000 µg/kg in rats in an acute 3 day administration to detect if the treatment with raloxifene produced acute changes in the hemostatic markers [18-21].

  1. Visvanathan, K.; Fabian, C. J.; Bantug, E.; Brewster, A. M.; Davidson, N. E.; Decensi, A.; Floyd, J. D.; Garber, J. E.; Hofstatter, E. W.; Khan, S. A.; Katapodi, M. C.; Pruthi, S.; Raab, R.; Runowicz, C. D.; Somerfield, M. R. Use of Endocrine Therapy for Breast Cancer Risk Reduction: ASCO Clinical Practice Guideline Update. Journal of Clinical Oncology 2019, 37(33): 3152–3165.
  2. Qaseem, A.; Hicks, L. A.; Etxeandia-Ikobaltzeta, I.; Shamliyan, T.; Cooney, T. G.; Clinical Guidelines Committee of the American College of Physicians; Cross, J. T.; Jr, Fitterman, N.; Lin, J. S.; Maroto, M.; Obley, A. J.; Tice, J. A.; Tufte, J. E. Pharmacologic Treatment of Primary Osteoporosis or Low Bone Mass to Prevent Fractures in Adults: A Living Clinical Guideline From the American College of Physicians. Annals of internal medicine 2023, 176(2): 224–238.
  3. Wishart, D. S.; Feunang, Y. D.; Guo, A. C.; Lo, E. J.; Marcu, A.; Grant, J. R.; Sajed, T.; Johnson, D.; Li, C.; Sayeeda, Z.; Assempour, N.; Iynkkaran, I.; Liu, Y.; Maciejewski, A.; Gale, N.; Wilson, A.; Chin, L.; Cummings, R.; Le, D.; Pon, A.; et al. DrugBank 5.0: a major update to the DrugBank database for 2018. Nucleic acids research 2018, 46(D1): D1074–D1082.
  4. Felgel-Farnholz, V.; Hlusicka, E. B.; Edemann-Callesen, H.; Garthe, A.; Winter, C.; Hadar, R. Adolescent raloxifene treatment in females prevents cognitive deficits in a neurodevelopmental rodent model of schizophrenia. Behavioural brain research 2023, 441, 114276.

Meanwhile, the authors should analyze in the Discussion section whether the changes in the relevant indicators are intrinsically linked between the Rx and E2 treatment groups and why; furthermore, the magnitude of the change in dose 10, 100 seems to be more significant in the changes in some indicators (factor XI and factor VII activity) than in dose 1000, and why this is happened if there is no absolute dose-dependent linear relationship in the changes in these indicators along with doses, these are missing in the manuscript and need to be analyzed in the Discussion section.

A: thank you for your observation, we believe we can enrich our work by addressing this concern. In our experience, sex hormones in general have a dual behavior dependent on the concentration, this is the case for estrogens as well, and the effect can be dependent on the tissue used, usually this observing a “U” shaped response, we will add the following text to the manuscript to illustrated this behavior and the following references:

L339 “It is important to note that the exact hemostatic effects of raloxifene may vary de-pending on the dosage, duration of treatment, and individual characteristics of the ex-perimental models. Interestingly, estrogens, as do sex hormones in general, have a bi-phasic behavior dependent of the concentration used.”

Montaño, L. M.; Calixto, E.; Figueroa, A.; Flores-Soto, E.; Carbajal, V.; Perusquía, M. Relaxation of androgens on rat thoracic aorta: testosterone concentration dependent agonist/antagonist L-type Ca2+ channel activity, and 5beta-dihydrotestosterone restricted to L-type Ca2+ channel blockade. Endocrinology 2008, 149(5): 2517–2526.

Almstrup, K.; Fernández, M. F.; Petersen, J. H.; Olea, N.; Skakkebaek, N. E.; Leffers, H. Dual effects of phytoestrogens result in u-shaped dose-response curves. Environmental health perspectives 2002, 110(8): 743–748.

Townsend, E. A.; Thompson, M. A.; Pabelick, C. M.; Prakash, Y. S. Rapid effects of estrogen on intracellular Ca2+ regulation in human airway smooth muscle. American journal of physiology. Lung cellular and molecular physiology 2010, 298(4): L521–L530.

Minors:

L11/L14: ‘1 to 1000 μg/Kg’, the specific experimental dose needs to be shown here, rather than using a concentration range.

A: Thank you for your comment, we have change this and listed the doses on the specified lines.

L17: ‘1-100’, it means from 1 to 100, but only 1 and 100 dose were detected in this study, so ‘1, 100’ would be more accurate.

A: Thank you for your observation, although the 4 concentrations were used for this experiment, in L17 we are indicating the concentrations that diminished F XI, which are 1, 10 and 100, we had decided to place it as a range to save on words due to the limited number of words allowed. We have changed the text in the manuscript to indicate the 3 concentrations that showed this effect.

L50: APC, The full name is needed when the abbreviation is first shown.

A: Thank you for the correction, we have included the full word of the abbreviation.

L50: activated protein C (APC)

L63: ‘a the’, please check the grammar.

A: Thank you for your observation, we have made the correction and will revise the full manuscript.

L105: ‘lab diet 5001.’ What’s the mean 5001 here, please check and revise it.

A: Thank you for the inquiry, 5001 is the name brand of the rodent lab diet that our institute uses, we have attached a link of the university of the food https://di.facmed.unam.mx/comisiones/Composici%C3%B3n%20del%20alimento%20Laboratory%20Rodent%20Diet%205001.pdf.

L201: ‘as percentage ± standard error of the mean (SEM)’, In the context, there was presented as mean ± SEM (L228, L243). Please check and correct it.

A: We are grateful for your comments, Figure 1 uses mean and Figures 2 and 3 use percentage, therefore we have added this to the text and made the corresponding changes in the figure legends to

L227,242, 263: ‘Raloxifene an estradiol’, ‘an’ or ‘and’, please check and revise them.

A: We thank the reviewer for his advice, we have corrected the error in the text and will revise the full manuscript.

L315-317, ‘Specifically, the highest dose of Raloxifene resulted in an increase in prothrombin time and a significant inhibitory effect in factor VII activity’. Actually, for the inhibitory effect of Raloxifene dose in factor VII activity, the dose 100 and 10 has the highest effects (Fig2A).

A: Thank you for your comments, we have added the following text in the manuscript

L320: Specifically, Raloxifene had an increase in prothrombin time (at the highest dose 1000 µg/kg) and significant inhibits the activity of factor VII (highest effect at 10 and 100 µg/kg), suggesting an increase in thrombotic risk.  

Figures:

Figures 2: Panel B, Check the panel letters in the Fig2 and Fig3. 

A: Thank you for your observation, we have revised all of the figures and adjusted the panel lettering.

L263, ‘Figure 3. Raloxifene an estradiol on activity of coagulation factor VII (A)’, In y axis of panel Fig3A, it was factor XI. Please check it.

A: Thank you for the generous comments, we have corrected the text in figure legend.

Reviewer 3 Report

Title: Hemostatic effects of raloxifene in ovariectomized rats

Keywords: raloxifene; 17β-Estradiol; hemostasis; ovariectomized rats; thrombosis

There are repeated keywords in the text. Authors must choose only one location. Do not use repeated words in both. Since the title is also indexed.

The authors could improve this: “Conclusions 348

This present study supports evidence the hemostatic effects of raloxifene in ovariec- 349

tomized rats. The results demonstrated ...”- obviously the conclusions are from this study!

“… This effect of raloxifene was caused by regulating the activity of coagulation factors 353

VII and XI, while increasing fibrinogen concentration. These findings suggest that raloxi- 354 fene modify coagulation parameters, decrease the risk of thrombosis in ovariectomized 355

rats thereby showcasing its potential as a therapeutic option for conditions such as osteo- 356 porosis, especially in patients with a risk of venous thrombosis. It is important to note that 357 the exact hemostatic effects of raloxifene may vary depending on the dosage, duration of 358 treatment, and individual characteristics of the experimental models. ….”   Wouldn't that look better in the discussion?

The conclusion should be simple and robust. Not showing results. Finish on top of them.

The references are quite old. Only 3 works greater than 2016. This could be updated.

Author Response

Reviewer 3

Comments and Suggestions for Authors

Title: Hemostatic effects of raloxifene in ovariectomized rats

Keywords: raloxifene; 17β-Estradiol; hemostasis; ovariectomized rats; thrombosis

There are repeated keywords in the text. Authors must choose only one location. Do not use repeated words in both. Since the title is also indexed.

A: Thank you for your comments, we have changed some keywords.

L21: coagulation

The authors could improve this: “Conclusions 348

This present study supports evidence the hemostatic effects of raloxifene in ovariec- 349

tomized rats. The results demonstrated ...”- obviously the conclusions are from this study!

“… This effect of raloxifene was caused by regulating the activity of coagulation factors 353

VII and XI, while increasing fibrinogen concentration. These findings suggest that raloxi- 354 fene modify coagulation parameters, decrease the risk of thrombosis in ovariectomized 355

rats thereby showcasing its potential as a therapeutic option for conditions such as osteo- 356 porosis, especially in patients with a risk of venous thrombosis. It is important to note that 357 the exact hemostatic effects of raloxifene may vary depending on the dosage, duration of 358 treatment, and individual characteristics of the experimental models. ….”   Wouldn't that look better in the discussion?

The conclusion should be simple and robust. Not showing results. Finish on top of them.

A: Thank you for your fine comment, we have modified the conclusion and moved some of the mentioned text to the discussion. We have added the following text:

L358-361: This study provides evidence that raloxifene has hemostatic effects on ovariectomized rats by influencing key factors in both the extrinsic and intrinsic pathways of the coagulation system. However, further studies are necessary to elucidate the underlying mechanisms of Rx's hemostatic effects and to evaluate its safety profile in human subjects.

L280-295: as observed by the significant increment in both the prothrombin time and activated partial thromboplastin time. This effect of raloxifene was caused by regulating the activity of coagulation factors VII and XI, while increasing fibrinogen concentration. Raloxifene modifies the coagulation parameters, decreases the risk of thrombosis in ovariectomized rats thereby showcasing its potential as a therapeutic option for conditions such as osteoporosis, especially in patients with a risk of venous thrombosis.

The references are quite old. Only 3 works greater than 2016. This could be updated.

A: Thank you for your fine comment, we added references.

Authors are grateful to the Editor and Referees for their comments and observations and are hopeful that with the amendments described herein our manuscript might be publishable in your distinguished journal.

Reviewer 4 Report

The problem described in the manuscript is interesting, the aim of the study is clear, experiments were planned and carried out properly.

The results suggested that raloxifene modify coagulation parameters and probably may decrease the risk of thrombosis in ovariectomized
rats. Authors suggest that these results showed its potential as a therapeutic model for osteoporosis, psrticularly in patients with a risk of venous thrombosis. The  hemostatic effects of raloxifene  dependent on the dosage, duration of treatment and probably the animal model. So, the conclusions should be more cautious particularly in comparison with wome. However, in spite of interesting problem, the manuscript should quote more recent publications which hopefully help to elucidate the mechanism of raloxifene action.

Author Response

July 7, 2023

Guest Editor

Pharmaceutical Science

Life Editorial Office

Herein you will find our article Title (life-2470867): Hemostatic effects of raloxifene in ovariectomized rats. Denys Alva-Chavarría, Maribel Soto-Núñez, Edgar Flores-Soto, and Ruth Jaimez*

We have addressed all the suggestions made by the reviewers and certainly/hope that our manuscript was consequently improved. All changes to the manuscript were highlighted in red and you will find the answers to each of the reviewers’ comments in the following pages. All their recommendations were fully met and a native speaker reviewed the language of the manuscript.

We hope you find our manuscript suitable for publication and look forward to hearing from you soon.

Thank you for your thoughtful consideration and receive my kindest regards.

Cordially,

Ruth Jaimez, Ph. D.

Pharmacology Department

Universidad Nacional Autónoma de México

Phone +52 555 623 2163

e-mail: jaimezruth@hotmail.com

Reviewer 4

The problem described in the manuscript is interesting, the aim of the study is clear, experiments were planned and carried out properly.

The results suggested that raloxifene modify coagulation parameters and probably may decrease the risk of thrombosis in ovariectomized
rats. Authors suggest that these results showed its potential as a therapeutic model for osteoporosis, particularly in patients with a risk of venous thrombosis. The  hemostatic effects of raloxifene  dependent on the dosage, duration of treatment and probably the animal model. So, the conclusions should be more cautious particularly in comparison with women. However, in spite of interesting problem, the manuscript should quote more recent publications which hopefully help to elucidate the mechanism of raloxifene action.

A: Thank you for your fine comment, we have modified the conclusion. We have added the following text:

L358-361: This study provides evidence that raloxifene has hemostatic effects on ovariectomized rats by influencing key factors in both the extrinsic and intrinsic pathways of the coagulation system. However, further studies are necessary to elucidate the underlying mechanisms of Rx's hemostatic effects and to evaluate its safety profile in human subjects.

A: Thank you for your fine comment, we added some references, although the references specifically of raloxifene in the coagulation system are limited and few are recent, which we have added, although we could not find anything pertaining to mechanisms of action.

Authors are grateful to the Editor and Referees for their comments and observations and are hopeful that with the amendments described herein our manuscript might be publishable in your distinguished journal.

Round 2

Reviewer 1 Report

I recommend the rejection of the manuscript titled: Hemostatic effects of raloxifene in ovariectomized rats, sheds light on topic of effects of raloxifene (Rx) and estradiol (E2) on hemostatic parameters of experimental rats.  The evaluation is based on that overall value of data presented and novelty of the idea in addition to the experimental design and quality of data presented in this manuscript writing. In addition, it is on the scope of this outstanding journal. However, the number of rats in this study is very low which limits the findings of this investigation reduces the quality of this current manuscript.The package of data presented is also very low to get significant findings out of this investigation. Actually, there is no real change in the revised manuscript from the old version.

Author Response

July 13, 2023

Guest Editor

Pharmaceutical Science

Life Editorial Office

Herein you will find our article Title (life-2470867): Hemostatic effects of raloxifene in ovariectomized rats. Denys Alva-Chavarría, Maribel Soto-Núñez, Edgar Flores-Soto, and Ruth Jaimez*

We have addressed all the suggestions made by the reviewers and certainly/hope that our manuscript was consequently improved. All changes to the manuscript were highlighted in red and you will find the answers to each of the reviewers’ comments in the following pages. All their recommendations were fully met and a native speaker reviewed the language of the manuscript.

We hope you find our manuscript suitable for publication and look forward to hearing from you soon.

Thank you for your thoughtful consideration and receive my kindest regards.

Cordially,

Ruth Jaimez, Ph. D.

Pharmacology Department

Universidad Nacional Autónoma de México

Phone +52 555 623 2163

e-mail: jaimezruth@hotmail.com

Reviewer 1

Although the authors didn't clearly explain how many animals were used ("4 rats per box" in the Animals section) and ("rats were distributed into nine groups" in the experimental design section). Please clearly explain how many animals were used and how they were divided only in the "animals section", while explain what they received in the "rats were distributed into nine groups". I believe that the number used is sufficiently enough.

A: We are sorry for the confusion, in the Section Animals L105 that states “4 rats per box” we are not indicating the number of the rats per experiment, only explaining the living conditions in which the rats were placed, that due to the dimensions of the enclosures, only 4 rats fit per box.

In the Section Experimental design L119 “distributed into nine groups” were the living conditions during the duration of the experiment, the rats can only be kept up to 4 rats per box due to ethical reason, and were distributed into either the control, raloxifene or estradiol groups. For each dosage the n=15 in each group, therefore, at the end of the experiment the n=60 for the group of raloxifene, n=60 for the estradiol group and n=15 for the control group.

The following text was added to the manuscript

L119 A total of 135 animal were used for the experiment

L120 due to the enclosure capacities the experiments could only be carried out by nine groups at a time

L123 n=15

L124 total n=60;

L129 total n=60;

Round 3

Reviewer 1 Report

The manuscript titled: Hemostatic effects of raloxifene in ovariectomized rats, sheds light on topic of effects of raloxifene (Rx) and estradiol (E2) on hemostatic parameters of experimental rats which represents a crucial in animal biology and hematology. Indeed, the idea of the manuscript is novel and it is well written using the standard format of life journal.

Although the authors included some data that increased the number of rats used in this study. However, the number of rats in this study is not enough to get real findings out of this investigation and subsequently reduces the quality of this current manuscript. In addition, the package of data presented is  low to get significant findings out of this investigation. Therefore, i recommend rejection of this manuscript.